# A Mechanistic Model for Simulation of Carbendazim and Chlorothalonil Transport through a Two-Stage Vertical Flow Constructed Wetland

Stan Wehbe [1,*], Feleke Zewge [2], Yoshihiko Inagaki [3], Wolfram Sievert [4], Tirumala Uday Kumar Nutakki [5] and Akshay Deshpande [6]

1   Water Science & Technology, African Center of Excellence for Water Management,
    Addis Ababa P.O. Box 1176, Ethiopia
2   Department of Chemistry, Faculty of Science, Addis Ababa University, Addis Ababa P.O. Box 1176, Ethiopia;
    zewge@chem.aau.edu.et
3   Department of Civil and Environmental Engineering, Waseda University, 3-4-1 Okubo, Shinjuku,
    Tokyo 169-8555, Japan; w-inagaki@ruri.waseda.jp
4   Sievert Consult, Schuhstraße 15, 32657 Lemgo, Germany; wolfram@sievertconsult.de
5   RAK Research and Innovation Centre, American University of Ras Al Khaimah,
    Ras Al Khaimah P.O. Box 31208, United Arab Emirates; uday.kumar@aurak.ac.ae
6   Department of Research, Reed Bed Wastewater Treatment, Dubai, United Arab Emirates; research@reedbed.ae
*   Correspondence: stanislas.vincent@aau.edu.et or stanwehbe@gmail.com

**Abstract:** A mechanistic model was developed to simulate one-dimensional pesticide transport in two-stage vertical flow constructed wetland. The two pesticides taken under study were carbendazim and chlorothalonil. The water flow patterns within the constructed wetland were simulated using the Richards equation. Water content and vertical flux, which are the outputs of the substrate water flow model, were used to calculate the substrate moisture-related parameters and advection term in the solute transport model. The governing solute transport equation took into account a total of six processes: advection, molecular diffusion, dispersion, adsorption to the solid surface, degradation and volatilization. A total of 14 simulation cases, corresponding with available experimental data, were used to calibrate the model, followed by further simulations with standardized influent pesticide concentrations. The simulations indicated that the constructed wetland reached a steady state of pesticide removal after 7 days of operation. Two distinct water flow patterns emerged under saturated and unsaturated conditions. The patterns observed while varying the hydraulic loading rates were similar for each individual saturation condition. Two-factor ANOVA of the simulated data further revealed that the carbendazim and chlorothalonil removal was dependent on the hydraulic loading rates, but it was independent of the influent pesticide concentration. Analysis of the simulated pesticide removal showed that degradation emerged as the predominant removal process over time for both the pesticides. The model developed in this study can be an important tool for the design and construction of treatment wetlands for pesticide removal from wastewater.

**Keywords:** Ethiopian floriculture industry; mechanistic model; pesticide removal; solute transport; two-stage vertical flow constructed wetland

## 1. Introduction

The floriculture industry has become a vital part of the Ethiopian economy, generating a revenue of $541 million from the export of cut flowers between 2021 and 2022. It was the third-largest source of forex income for Ethiopia in 2022 after coffee and gold exports. Ethiopia has also emerged as the second-largest exporter of cut flowers in Africa and the fifth-largest exporter of cut flowers in the world [1,2]. A recent study assessing the performance of the Ethiopian floriculture industry described it as having a relatively competitive and consistent performance trend in the global market [3]. The increasing

exports of cut flowers over the years come with an increased usage of pesticides which support an increased scale of flower production.

Floriculture farms are often situated close to water bodies [4]. Water runoffs from these farms often find their way into these water bodies, and with accumulation over time, they have the potential to harm aquatic life and humans living around these water bodies. There are two registered pesticide products in particular that are used in floriculture farms which contain the pesticides carbendazim and chlorothalonil. Carbendazim has a half-life of anywhere between 3 days to 12 months while chlorothalonil has been shown to have a half-life of 8 days in water [5,6]. Both of these chemicals have already been banned in the European Union [7,8].

One of the ways to treat pesticide-containing wastewater is the use of constructed wetlands (CWs). Pesticide removal using CWs has been studied widely since the 1970s [9]. The Ethiopian Horticulture Producer Exporters Association has also made efforts to support the construction of CWs in floriculture farms [10]; however, there is room for better design and implementation strategies [11].

There are various types of CWs that have been studied over the years for the treatment of agricultural runoffs [12,13]. The use of vertical flow constructed wetlands for pesticide treatment has been studied recently [14–17], which has shown great promise towards efficient removal of pesticides. A lot of these studies usually describe the removal efficiency of the pesticides from wastewater; however, very few manage to conduct a more detailed study of pesticide concentration dynamics as the wastewater passes through the substrate bed [18].

One of the main challenges faced in evaluating pesticide transport through the substrate bed is the testing of a large number of core samples needed for such an evaluation. Testing for pesticide residues needs a mass spectrometer, which makes testing of a large number of samples resource-intensive. Furthermore, in the case of field studies, there is a need to sample from different parts of the CW to obtain a representative sample. Thus, larger sample quantities make the process laborious and time-consuming as well [19].

For this reason, it is advantageous to develop predictive models for pesticide transport and substrate water flow to evaluate the CW performance as a whole. Predictive models can help appraise the performance of proposed CW designs over time and optimize several parameters in the design stage prior to actual construction. These models can also serve as tools for inspections during environmental impact assessments of constructed CWs.

Simple models (e.g., a regression model or lumped deterministic model), although they are often used for the design of CW, provide a limited understanding of the system due to the simplified formulation of the system. A mechanistic approach, on the other hand, can simulate spatial and temporal variations of pesticide concentrations inside the CW. Since the vertical flow CW modeling needs to deal with a variably saturated condition, the implementation is more complex compared to that for saturated conditions [20]. In response to this situation, the numerical implementation has been advanced considerably in the last decade [21]. CW2D modules in HYDRUS and CWM1 embedded in the OpenGeosys are popular among vertical flow CW models and have been used for various purposes such as evaluations of intensified CW [22,23] and clogging [24]. Two- or three-dimensional computational fluid dynamics models have recently emerged as a new mechanistic CW simulation tool [25–28]. They reveal very detailed flow characteristics as well as pollutant behaviors inside the system; however, most of the studies are for conventional water quality variables and horizontal surface flow CW. Overall, the application of the mechanistic model is still in its infancy for vertical flow CW pesticide studies.

## 2. Model Description and Simulation Methods

### 2.1. Model Description

The model developed in this study was a process-based one developed on the basis of previous work by Gottardi and Venutelli [29] to develop a simulation of one-dimensional vertical flow of pesticide-containing wastewater through a CW under saturated and unsat-

urated conditions. Additionally, the fate of the pesticide as it passes through the sand and gravel layers was also simulated.

The specific design of the CW followed was adapted from the previous experimental work in Wehbe et al. (2022) and Wehbe et al. (2023) [16,17]. Substrate water flow was described with the Richards equation, while the movement of pesticide was described with an advection–dispersion equation coupled with diverse processes, which are detailed in the later Section 2.3.

### 2.2. Substrate Water Flow

One-dimensional vertical flow of water for unsaturated conditions was described using the Richards equation [29]:

$$\frac{\partial \Theta}{\partial t} = \frac{\partial}{\partial z}\left[K\left(\frac{\partial h}{\partial z} - 1\right)\right] \tag{1}$$

where $\Theta$ is the volumetric water content, $h$ is the pressure head, $K$ is the unsaturated hydraulic conductivity, $z$ is the vertical coordinate in the downward direction from the top surface, $t$ is the time. $\theta$ and $K$, which are functions of $h$, can be represented using the van Genuchten–Mualem functions:

$$\theta(h) = \theta_r + \frac{\theta_s - \theta_r}{[1 + |\alpha h|^n]^m}\,(h < 0); \theta_s\,(h \geq 0) \tag{2}$$

$$K(h) = K_s S_e^{0.5}\left[1 - \left(1 - S_e^{1/m}\right)^m\right]^2 (h < 0); K(h) = K_S\,(h \geq 0) \tag{3}$$

$$S_e = \frac{\theta - \theta_r}{\theta_s - \theta_r}$$

where $\theta(h)$ is the volumetric water content, $\theta_r$ and $\theta_s$ are residual and saturated water contents, respectively, $K(h)$ is the unsaturated hydraulic conductivity, $K_s$ is the saturated hydraulic conductivity; $\alpha$, $n$ and $m$ are empirical shape parameters [$n = 1/(1 - m)$], $S_e$ is the effective volumetric water content.

Initial conditions for the pressure head within the system domain were specified as follows:

$h_0(z)$ is the initial water pressure head at different soil depths, and $L$ is the maximum CW depth (cm). The boundary conditions at the CW surface ($z = 0$) or at the bottom ($z = L$) are expressed as specified flux and specified pressure head.

$$K\frac{\partial h}{\partial z} + K = q_0(t) \tag{4}$$

$$h(L, t) = 0 \tag{5}$$

where $q_0(t)$ is a water flux due to feeding at the upper boundary.

The initial condition for pressure head within the system domain was set at: $h\,(z, t = 0) = -65$ (cm) for the unsaturated condition; $h\,(z, t = 0) = -0.1$ (cm) ($z < z_{water}$) and a hydrostatic distribution of pressure head [$h(z,t = 0) = z - z_{water}$ ($z \geq z_{water}$)] for the saturated condition, where $z_{water}$ (=5 cm) is the position of the water table from the top.

### 2.3. Solute Transport

The model takes into account the following processes: (i) advection; (ii) molecular diffusion; (iii) dispersion; (iv) adsorption to the solid surface; (v) degradation; (vi) volatilization [30]. Root uptake was excluded, as the evapotranspiration was negligibly small; however, the effect of plants (e.g., activation of microorganisms due to the presence of plant roots and facilitation of adsorption of contaminants) can be evaluated from degradation and adsorption terms.

Substrate water flow and solute transport were coupled: water content and vertical flux, which are the outputs of the soil water flow model, were used to calculate the soil moisture-related parameters and advection term in the solute transport model.

The governing equation for one-dimensional solute transport is described as the following [30] :

$$\frac{\partial C}{\partial t} = \frac{1}{\theta + \rho K_d + \alpha K_H} \left\{ \frac{\partial}{\partial z} \left[ \alpha D_g \frac{\partial (K_H C)}{\partial z} \right] + \frac{\partial}{\partial z} \left[ \theta D_l \frac{\partial C}{\partial z} \right] - \frac{\partial (qC)}{\partial z} - (\theta + \rho K_d + \alpha K_H) k_s C \right\} \tag{6}$$

where $C$ is the solute concentration in the liquid phase. $q$ represents volumetric flux (or hydraulic load). $\rho$ is the bulk density. $k$ is a first-order degradation rate constant. $D$ is the dispersion coefficient, which accounts for both the molecular diffusion and the mechanistic dispersion:

$$D_l = \left( \frac{\theta^{\frac{7}{3}}}{n^2} \right) \cdot D_l{}^w + \alpha_L \cdot \frac{|q|}{\theta} \tag{7}$$

where $n$ is a porosity, $D_l{}^w$ is the molecular diffusion coefficient in water (cm$^2$ d$^{-1}$), $\alpha_L$ is the longitudinal dispersivity (cm).

As an initial condition, pesticide concentrations were assumed to be 0 in the entire system: $C(z, t = 0) = 0$. The boundary conditions at the CW surface ($z = 0$) or at the bottom ($z = L$) can be expressed as:

$$-E^T \frac{\partial C}{\partial z} + qC = qC_{in} - \frac{D_g K_H}{d} C \tag{8}$$

$$-E^T \frac{\partial C}{\partial z} + qC = qC_{out} \tag{9}$$

where,

$$E^T = \alpha D_g K_H + \theta D^l \tag{10}$$

where, $d$ is the thickness of the air boundary layer. $C_{out}$ can be equal to $C$, as $\partial C / \partial z = 0$ at the bottom.

Under the saturated condition, it was assumed that solute transport associated with advection occurs only during the flush time.

*2.4. Numerical Calculation Methods*

In this study, implicit Euler temporal discretization and cell-centered finite-difference spatial discretization were applied. The h-based Richards equation can be solved with the finite-difference approximation coupled with a modified Picard iteration method [31]:

$$\begin{aligned} C_i{}^m \frac{h_i{}^{n+1,m+1} - h_i{}^n}{\Delta t} &= \frac{K_{i-1/2}{}^m}{(\Delta z)^2} \left( h_{i-1}{}^{m+1} - h_i{}^{m+1} \right) \\ &+ \frac{K_{i+1/2}{}^m}{(\Delta z)^2} \left( h_{i+1}{}^{m+1} - h_i{}^{m+1} \right) - \frac{K_{i+1/2}{}^m - K_{i-1/2}{}^m}{\Delta z} \end{aligned} \tag{11}$$

where $C$ is hydraulic capacity, superscripts, $n$ and $m$, refer to time and iteration levels, respectively, and $\Delta t$ is the time step. Here, the pressure head at the mth iteration denoted as $h^m$ is related with a truncated Taylor series, i.e., the $h_i{}^{n+1,m+1}$ is expressed with respect to $h$:

$$h_i{}^{n+1,m+1} = h_i{}^{n+1,m} + \delta_i{}^m = h_i{}^{n+1,m} + \left( h_i{}^{n+1,m+1} - h_i{}^{n+1,m} \right) \tag{12}$$

The iteration scheme is expressed as

$$A_i h_{i-1}{}^{j+1,m+1} + B_i h_i{}^{j+1,m+1} + C_i h_{i+1}{}^{j+1,m+1} = R_i \tag{13}$$

$A_i$, $B_i$, $C_i$, $R_i$ are matrix coefficients at the mth iteration level equation.

$$A_i = \frac{K_{i-1/2}{}^m}{(\Delta z)^2}, \ B_i = \frac{C_i{}^m}{\Delta t} + \frac{K_{i-1/2}{}^m}{(\Delta z)^2} + \frac{K_{i+1/2}{}^m}{(\Delta z)^2}, \ C_i = \frac{K_{i+1/2}{}^m}{(\Delta z)^2}$$

$$R_i = \frac{K_{i-1/2}{}^m}{(\Delta z)^2}\left(h_{i-1}{}^m - h_i{}^m\right) + \frac{K_{i+1/2}{}^m}{(\Delta z)^2}\left(h_{i+1}{}^m - h_i{}^m\right) - \frac{K_{i+1/2}{}^m - K_{i-1/2}{}^m}{\Delta z} - C_i{}^m \frac{h_i{}^m - h_i{}^n}{\Delta t} \tag{14}$$

The tridiagonal coefficient matrix can be solved using the Thomas' algorithm. The arithmetic mean of the hydraulic conductivity was calculated to obtain the hydraulic conductivity in the middle of two adjacent nodes. The scheme used to solve the solute transport equation was also implicit Euler temporal discretization and cell-centered finite-difference spatial discretization:

$$\frac{C_i{}^{T+1} - C_i{}^T}{\Delta t} = \frac{ELV}{R}\left(C_{i+1}{}^{T+1} - 2C_i{}^{T+1} + C_{i-1}{}^{T+1}\right) + \frac{ESL}{R}\left(C_{i+1}{}^{T+1} - 2C_i{}^{T+1} + C_{i-1}{}^{T+1}\right) - \frac{EQ}{R}\left(C_{i+1}{}^{T+1} - C_{i-1}{}^{T+1}\right) - k_s C_i{}^T \tag{15}$$

where $R = \Theta + \rho K_d + \alpha K_H$

$$ELV = \frac{\alpha D_g K_H}{(\Delta z)^2}, \ EQ = \frac{q}{2\Delta z}, \ ESL = \frac{\Theta D^l}{(\Delta z)^2}$$

The iteration scheme is expressed as

$$a_i C_{i-1}{}^{T+1} + b_i C_i{}^{T+1} + c_i C_{i+1}{}^{T+1} = r_i \tag{16}$$

$a_i$, $b_i$, $c_i$, and $r_i$ are matrix coefficients at the $T+1$ time level equation.

$$a_i = -\frac{ELV}{R}\Delta t - \frac{ESL}{R}\Delta t + \frac{EQ}{R}\Delta t \tag{17}$$

$$b_i = -\frac{2ELV}{R}\Delta t - \frac{2ESL}{R}\Delta t + k_s\Delta t + 1 \tag{18}$$

$$c_i = -\frac{ELV}{R}\Delta t - \frac{ESL}{R}\Delta t - \frac{EQ}{R}\Delta t \tag{19}$$

$$r_i = 1$$

The vertical nodal fluxes, $q_i$, were computed as the following:

$$q_1{}^{j+1} = -K_{1+1/2}{}^{j+1}\left(\frac{h_2{}^{j+1} - h_1{}^{j+1}}{\Delta z} - 1\right) \tag{20}$$

$$q_i{}^{j+1} = \frac{K_{i+1/2}{}^{j+1}\left(\frac{h_{i+1}{}^{j+1} - h_i{}^{j+1}}{\Delta z} - 1\right) - K_{i-1/2}{}^{j+1}\left(\frac{h_i{}^{j+1} - h_{i-1}{}^{j+1}}{\Delta z} - 1\right)}{2} \tag{21}$$

$$q_L{}^{j+1} = -K_{L+1/2}{}^{j+1}\left(\frac{h_L{}^{j+1} - h_{L-1}{}^{j+1}}{\Delta z} - 1\right) \tag{22}$$

For the stability of numerical calculations, the segment size and the maximum absolute ratio allowed for two subsequent iterations ($|h^{m+1} - h^m|/h^m$), varied depending on the saturation condition: they were set at 1 cm and $7 \times 10^{-4}$ for the saturated condition and 2 cm and $1.5 \times 10^{-16}$ for the unsaturated condition.

The specific parameters for carbendazim and chlorothalonil used in the simulation process are as listed in Table 1. The specific parameters of the substrate gravel used in the simulation process are as listed in Table 2. The computer code written in Fortran for the current study was extended from the work carried out by Gottardi and Venutelli [29].

**Table 1.** Simulation parameters for the pesticides carbendazim and chlorothalonil.

| Parameter | Unit | Carbendazim | | Chlorothalonil | |
|---|---|---|---|---|---|
| | | Value | Reference | Value | Reference |
| Henry Constant | - | $1.28 \times 10^{-6}$ | [8] | $8.06 \times 10^{-6}$ | [13] |
| Molecular diffusion coefficient in water | $cm^2 \, s^{-1}$ | $2.43 \times 10^{-6}$ | [18] | $1.39 \times 10^{-5}$ | [6] |
| Diffusion coefficient in air | $cm^2 \, s^{-1}$ | NA | [18] | $4.98 \times 10^{-2}$ | [6] |
| Organic carbon partition coefficient | $cm^2 \, g^{-1}$ | 450 | [8] | 3100 | [6] |
| First-order degradation rate constant | $s^{-1}$ | $7.29 \times 10^{-7}$ | [18] | $1.01 \times 10^{-6}$ (aerobic) $1.61 \times 10^{-5}$ (anaerobic) | [13] |

**Table 2.** Simulation parameters for the substrate [25].

| Parameter | Description | Unit | Value for Top Layer (Sand 1–2 mm) | Value for Middle and Bottom Layers (Gravel 2–10 mm, 10–20 mm) |
|---|---|---|---|---|
| $\theta_r$ | Residual water content | - | 0.075 | 0.04 |
| $\theta_s$ | Saturated water content | - | 0.37 | 0.43 |
| $\alpha$ | van Genuchten shape parameter | $cm^{-1}$ | 0.12246 | 0.18 |
| $n$ | | - | 2.8 | 3.3 |
| $K_s$ | Saturated hydraulic conductivity | $cm \, s^{-1}$ | 0.5155 | 1.1875 |

The distribution coefficients $K_d$ for carbendazim and chlorothalonil were calculated by multiplying the organic carbon partition coefficient ($K_{OC}$) with a fraction of organic carbon content ($f_{oc}$), which is assumed to be 0.12% [32]:

$$K_d = K_{OC} \times f_{OC} \tag{23}$$

*2.5. Contribution of Individual Processes*

To evaluate the contribution of relevant processes, an equation of temporal discretization and cell-centered finite-difference spatial discretization was exploited. The five processes were considered: *Disp* indicates hydrodynamic dispersion; *Flow* indicates advective transport and infiltration; *Decay* indicates biochemical degradation; *Evap* indicates evaporation; *Volat* is surface volatilization. Each process at a specific depth can be calculated as follows:

$$Disp(z) = \frac{ESL}{R}(C(z+dz)) - 2(C(z)) + C(z-dz) \tag{24}$$

$$Evap(z) = \frac{ESL}{R}(C(z+dz)) - 2(C(z)) + C(z-dz) \tag{25}$$

$$Flow(z) = \frac{EQ}{R}(C(z+dz)) - C(z-dz) \tag{26}$$

$$Decay(z) = k_s C(z) \tag{27}$$

$$Volat(z=0) = aD_g \times \frac{K_H}{Rd} \times C(z=0) \tag{28}$$

The mass balance equation for a single node can be described as the following:

$$\frac{\partial C}{\partial t} = Disp(z) + Evap(z) - Flow(z) - Decay(z) - Volat(z) \tag{29}$$

and it holds true for the summation form:

$$\sum_z^{Depth} \frac{\partial C(z)}{\partial t} = \sum_z^{Depth} Disp(z) + Evap(z) - Flow(z) - Decay(z) - Volat(z) \tag{30}$$

Each term on the right-hand side was calculated for every node according to the above equations, and the obtained values in each node were summed up per the individual process. Meanwhile, to evaluate the left-hand side, we calculated the difference of concentration levels at two different time points, dividing it by the time difference. For carbendazim, Evap and Volat were excluded due to their relatively non-volatile nature.

### 2.6. Methods of Simulation

The numerical calculations for the simulation were executed through Fortran programming language, while the data generated were visualized with the help of Python programming language. The simulation code files can be found in Supplementary Material Files S1 and S2. The simulations were conducted for a time period of 10 days.

#### 2.6.1. Simulation Cases

A total of 14 cases were considered, based on the available corresponding experimental data. The list of simulation cases is as described in Table 3. The details of the experimental setup are described in Wehbe et al. (2022) and Wehbe et al. (2023) [16,17]. There were two distinct cases: (i) carbendazim was treated under the unsaturated condition (corresponding to Case 7), and (ii) a very high concentration of chlorothalonil was fed to the system under the unsaturated condition (corresponding to Case 14) in terms of different natures of experiments for the treatment of each compound.

**Table 3.** A list of simulation cases.

| Case No. | Pesticide | Hydraulic Loading Rate | Influent Concentration | Water Saturation | Label |
|---|---|---|---|---|---|
| 1 | | $100 \text{ L/m}^3/\text{d}$ | $73 \text{ μg/L}$ | | C73-L100-S |
| 2 | | $200 \text{ L/m}^3/\text{d}$ | $43 \text{ μg/L}$ | | C43-L200-S |
| 3 | | $400 \text{ L/m}^3/\text{d}$ | $190 \text{ μg/L}$ | Saturated | C190-L400-S |
| 4 | Carbendazim | $100 \text{ L/m}^3/\text{d}$ | $100 \text{ μg/L}$ | | C100-L100-S |
| 5 | | $200 \text{ L/m}^3/\text{d}$ | $100 \text{ μg/L}$ | | C100-L200-S |
| 6 | | $400 \text{ L/m}^3/\text{d}$ | $100 \text{ μg/L}$ | | C100-L400-S |
| 7 | | $200 \text{ L/m}^3/\text{d}$ | $127 \text{ μg/L}$ | Unsaturated | C127-L200-U |
| 8 | | $50 \text{ L/m}^3/\text{d}$ | $100 \text{ μg/L}$ | | C76-L050-U |
| 9 | | $200 \text{ L/m}^3/\text{d}$ | $100 \text{ μg/L}$ | | C79-L200-U |
| 10 | | $400 \text{ L/m}^3/\text{d}$ | $100 \text{ μg/L}$ | | C72-L400-U |
| 11 | Chlorothalonil | $50 \text{ L/m}^3/\text{d}$ | $500 \text{ μg/L}$ | Unsaturated | C236-L050-U |
| 12 | | $200 \text{ L/m}^3/\text{d}$ | $500 \text{ μg/L}$ | | C171-L200-U |
| 13 | | $400 \text{ L/m}^3/\text{d}$ | $500 \text{ μg/L}$ | | C391-L400-U |
| 14 | | $50 \text{ L/m}^3/\text{d}$ | $50 \text{ mg/L}$ | | C050G-L050-U |

#### 2.6.2. Calibration of Parameters

Reported values for the model parameters were first used to reduce the complexity of model calibration. However, there were still deviations between the simulation outputs and observed experimental data. To minimize the absolute error between simulation outputs and measured concentrations, new parameters were introduced to the simulation calculations.

Lower HLRs could lead to reduced mass transfer of the solute to the substrate. To account for this, a new parameter of mass transfer resistance factor was introduced into the calculations as a multiplication factor to adjust the distribution coefficient ($K_d$) values. Lower HLRs could also lead to less uniform distribution of the pesticide in the CW, leading

to formation of dead zones in the CW where the degradation is limited. In order to account for this situation, a new parameter of degradation enhancement factor was introduced into the calculations as a multiplication factor to adjust the degradation rate constant ($K_s$).

For carbendazim, experimental data of effluent from Stage B were used for model calibration, while experimental data of effluent from Stage A were used in the case of chlorothalonil.

### 2.6.3. Simulation of Standardized Influent Pesticide Concentrations

In order to effectively compare the performance of the CW at different HLRs, there was a need to use uniform influent pesticide concentrations for simulations across all the different HLRs. To carry this out, carbendazim influent concentrations of 10 and 100 μg $L^{-1}$ and chlorothalonil influent concentrations of 100 and 500 μg $L^{-1}$ were chosen.

There was also a need to normalize the results to enable effective comparison of the results. The normalized factor used for carbendazim results was stage B effluent concentration divided by the influent concentration. The normalized factor used for chlorothalonil was stage A effluent concentration divided by the influent concentration.

Further analysis using two-factor ANOVA (at significance level of $p < 0.05$) was conducted to analyze how the influent concentrations and HLR affect the normalized factor values for both pesticides.

## 3. Results and Discussion

### 3.1. Substrate Water Flow Pattern

Water flow simulations showed a distinct difference between saturated and unsaturated conditions. The simulation data observed are illustrated in Figure 1. The simulation patterns observed under a particular saturation condition while varying the HLR were similar to each other. Such a trend is to be expected under saturated conditions; however, this trend repeated itself under unsaturated conditions as well. The cause of this trend could be the regular and consistent loading regime, of every four hours, which was followed for the purpose of this simulation.

As can be observed under both saturated and unsaturated conditions, there was a noticeable change at a depth of 40 cm in Stage A and at a depth of 60 cm in Stage B. This shift was due to the change at these particular depths in the two stages of the constructed wetland. The substrate grain size used in the top layer was 1–5 mm, while those used in the middle and bottom layers were 10–15 mm and 20–25 mm, respectively. The larger grain size of the substrate in the lower substrate layers allowed for increased water flow through these layers. For this reason, under unsaturated conditions, we observed a decrease in water content where the middle substrate layer of both stages begins, followed by an exponential increase in the water content with increasing depth. On the other hand, under saturated conditions, we observe a drastic increase in the volume of water where the middle substrate layer of stage A and stage B begins, followed by a constant volume of water with increasing depth.

Another point of note was that the water content profile of the CW under unsaturated conditions seemed to follow a consistent cycle with the loading regime. This observation is consistent with a previous study where the water content profile reached a pseudo-steady state after repeated flushes within a period of 12 h as well [33]. A representation of this cycle of water content is illustrated in Figure 2.

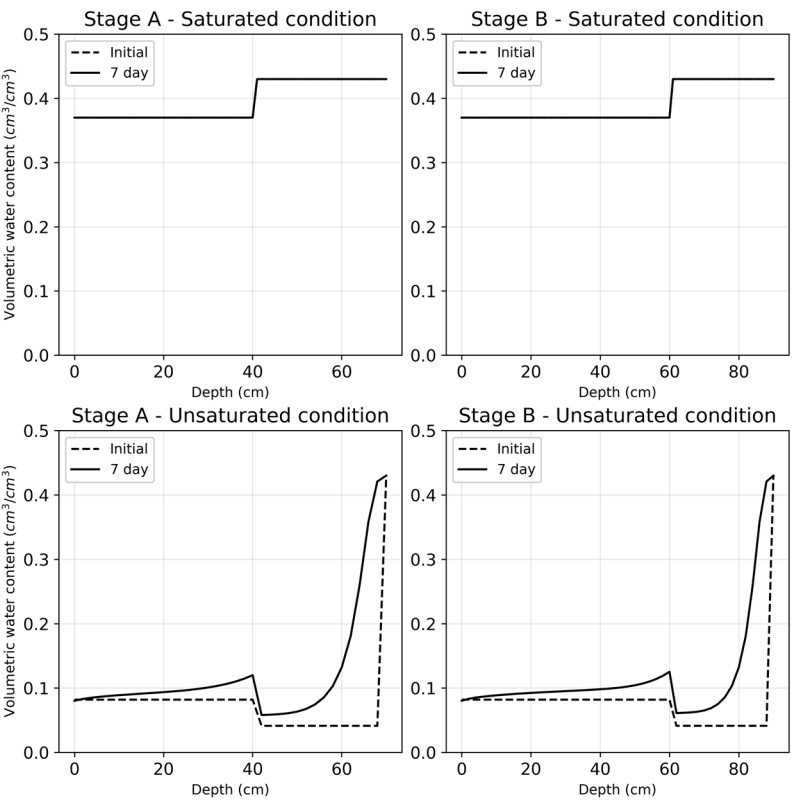

**Figure 1.** The four graph plots describe the simulated water flow patterns in stages A and B at different depths of the substrate bed under saturated and unsaturated conditions. The two plots at the top illustrate the water content under saturated conditions, while the bottom two graphs illustrate the water content under unsaturated conditions.

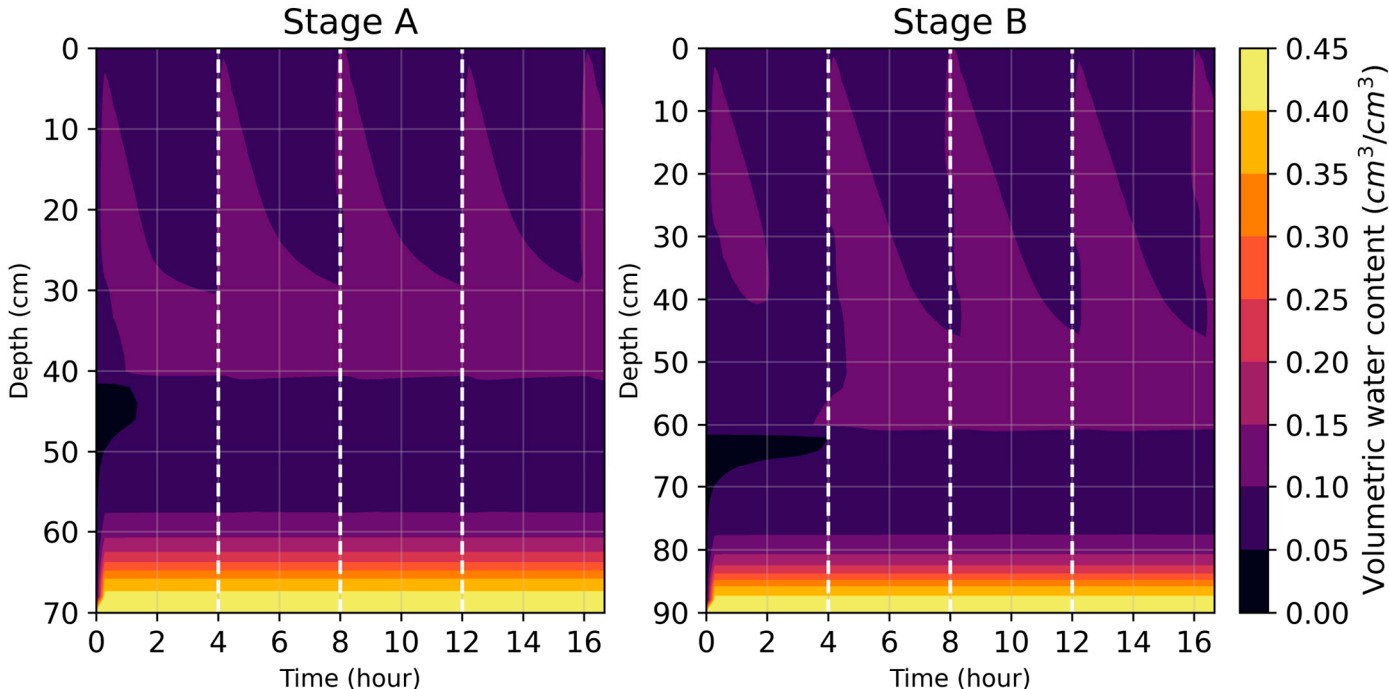

**Figure 2.** An illustration of the cyclic nature of the water content profile present in the CW, as observed from simulated data. A steady-state water content profile can be seen starting from the second loading regime.

### 3.2. Calibration of Simulation Parameters

Solute transport simulations were conducted using the data presented in Tables 1 and 2 for calculation purposes. The simulated effluent pesticide concentrations were compared with the experimental findings. From initial runs of the simulation, a lot of variation was observed between the experimental data and simulated data.

For carbendazim, the default $K_d$ value was set at 0.58, and the default $K_s$ value was set to $7 \times 10^{-7}$. The mass transfer resistance factor was set to 0.2 for HLR of 100 L d$^{-1}$, while it was set to 1 for HLR of 200 and 400 L d$^{-1}$. The degradation enhancement factor was set at 1 for HLR of 100 L d$^{-1}$, while it was set at 45 and 90 for HLRs of 200 and 400 L d$^{-1}$, respectively. For chlorothalonil, the default $K_d$ value was set at 6.5, and the default $K_s$ value was set to $1 \times 10^{-6}$. The mass transfer resistance factor was set to 0.4 and 0.6 for HLRs of 50 and 200 L d$^{-1}$, respectively, while it was set to 1 for HLR of 400 L d$^{-1}$. The degradation enhancement factor was set at 1 for all three HLRs of 50, 200 and 400 L d$^{-1}$. Note that calibration was not performed for cases for HLR of 400 L d$^{-1}$ for chlorothalonil so they were excluded from the table. A comparison of the calibrated simulation data with experimental data is illustrated in Figure 3, while a detailed summary of the factor values and errors can be found in Table S3.

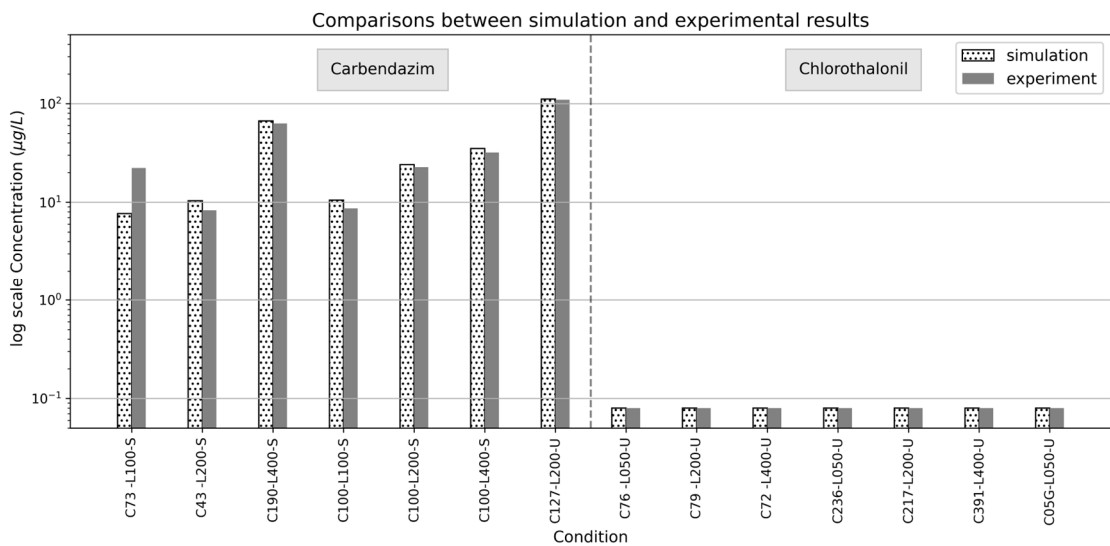

**Figure 3.** Comparison of simulated results with experimental data from previous studies for carbendazim and chlorothalonil removal using the same CW setup.

### 3.3. Simulation of Pesticide Transport through the CW

The simulations of pesticide solute transport were conducted for a period of 10 days. The simulated effluent concentration data for stages A and B over time are illustrated in Figure 4. It was observed that solute transport through the CW had either achieved or was close to achieving steady state after 7 days of operation for both carbendazim and chlorothalonil.

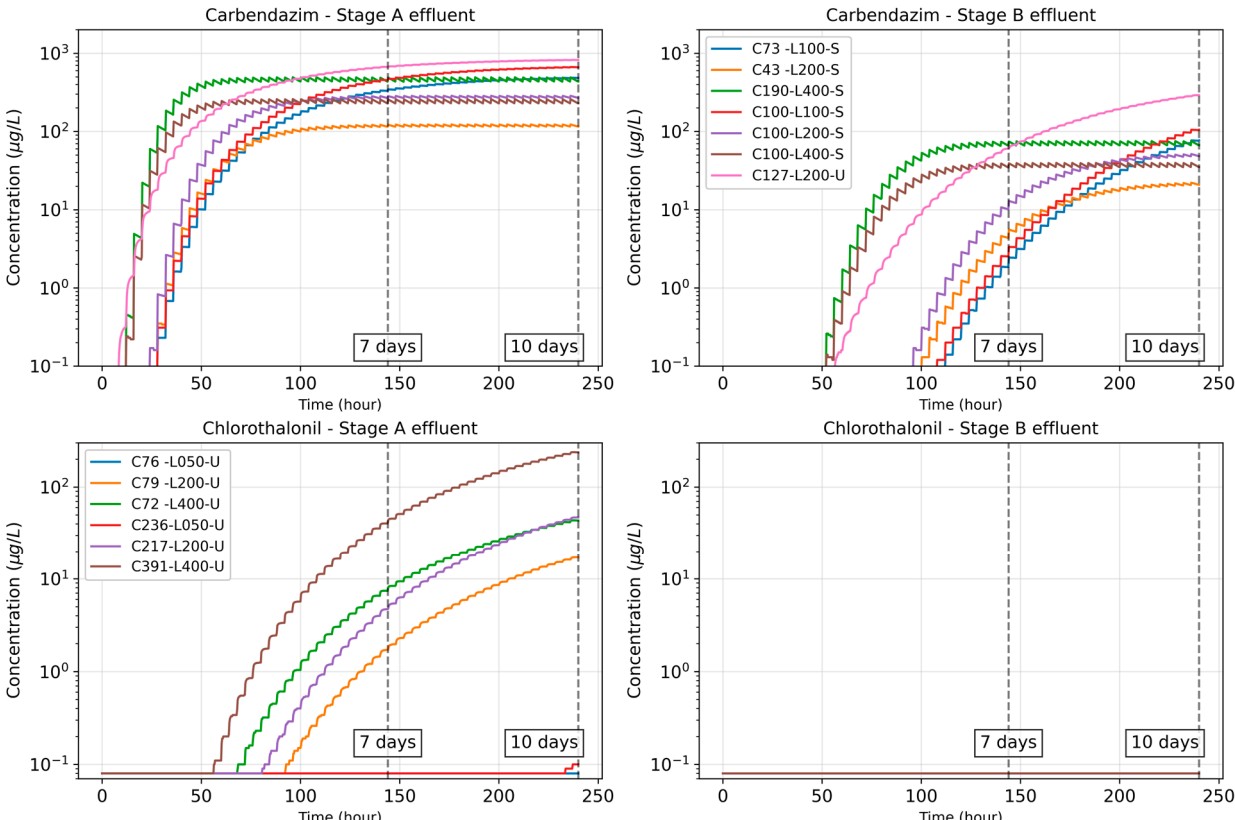

**Figure 4.** Illustration of pesticide effluent concentration data over a period of 10 days. Each individual line represents a particular simulation case. The colored lines for Carbendazim—Stage A effluent correspond with those of Carbendazim—Stage B effluent.

### 3.3.1. Carbendazim

The results of solute transport simulation for carbendazim, after 7 days of operation, were as illustrated in Figure 5. As can be observed, there is an accumulation of the carbendazim in the substrate bed over time. At a lower HLR of 50 L/d, stage A was sufficient to remove carbendazim from the influent water after one week of CW operation, while stage B was needed for effective carbendazim removal after one week of CW operation. The experimental findings for stage A carbendazim effluents were also similar to the findings for the calibrated simulation results.

### 3.3.2. Chlorothalonil

The results of solute transport simulation for chlorothalonil, after 7 days of operation, are as illustrated in Figure 6. Similar to the results from carbendazim simulation cases, there is an accumulation of the chlorothalonil in the substrate bed over time. At a lower HLR of 50 L/d, stage A was sufficient to remove chlorothalonil from the influent water after one week of CW operation, while stage B was needed for effective chlorothalonil removal after one week of CW operation. The experimental findings for stage A chlorothalonil effluents were also similar to the findings for the calibrated simulation results.

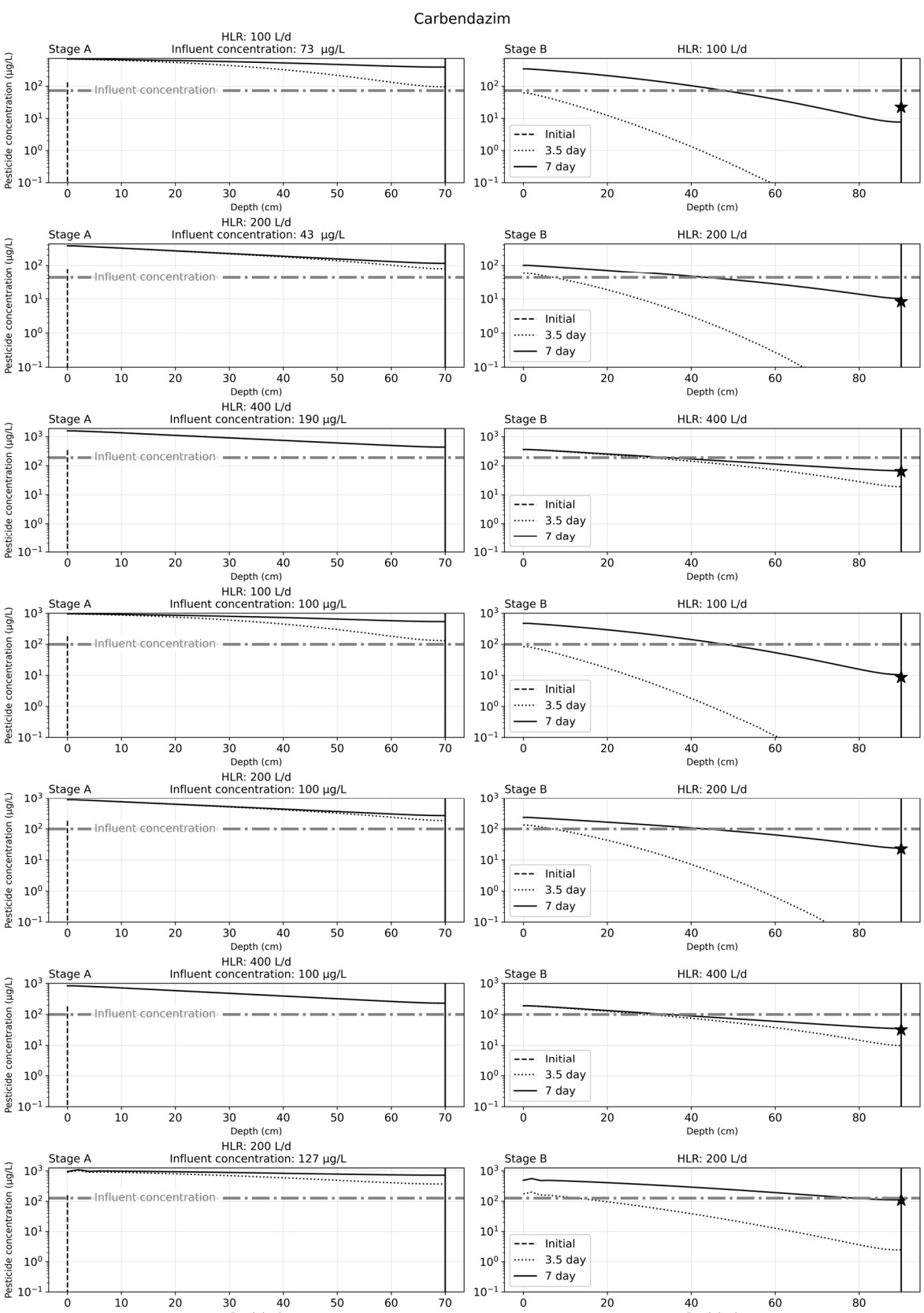

**Figure 5.** Solute transport results for the seven simulation cases used for carbendazim after 7 days of operation. The seven cases depicted in the figure correspond to findings from previous experimental work with carbendazim. The star in the graph for stage B indicates the effluent concentrations observed from experimental work.

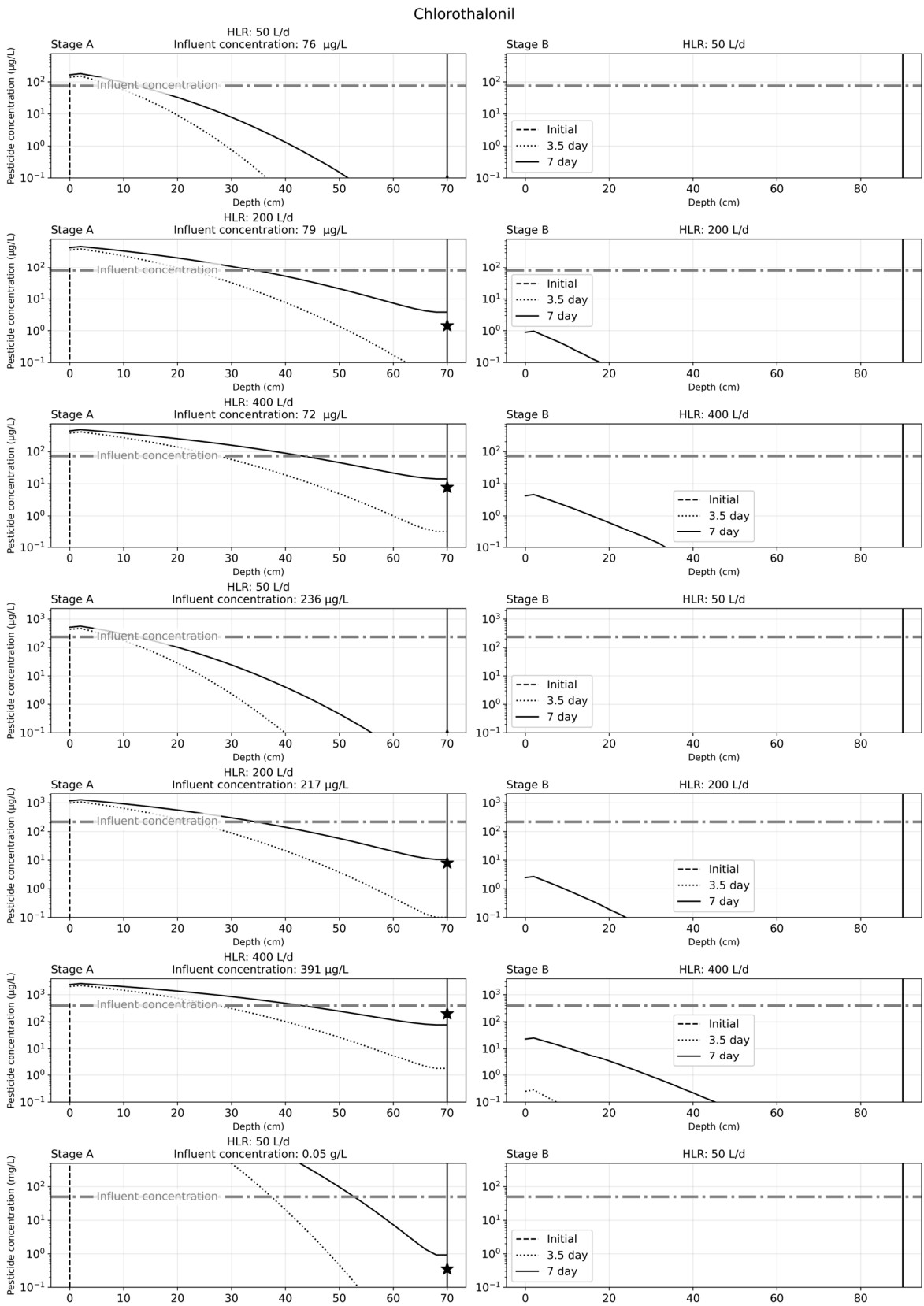

**Figure 6.** Solute transport results for the seven simulation cases used for chlorothalonil after 7 days of operation. The seven cases depicted in the figure correspond to findings from previous experimental work with chlorothalonil. The star in the graph for stage A indicates the effluent concentrations observed from experimental work.

*3.4. Contribution of Individual Processes in Pesticide Transport*

Figures 7 and 8 illustrate the contribution of different processes for selected cases (the other cases were omitted, since the results were fundamentally the same), just before and after a system flush. The process contributions were compared with the net decrease or increase in pesticide concentrations at a certain time point (characterized as dC/dt), which indicates the concentration change at a given time (i.e., 2, 4, 7 days). The negative dC/dt values indicate a net pesticide removal, while positive dC/dt values indicate a net pesticide accumulation. The removal or accumulation rate was balanced by considered processes.

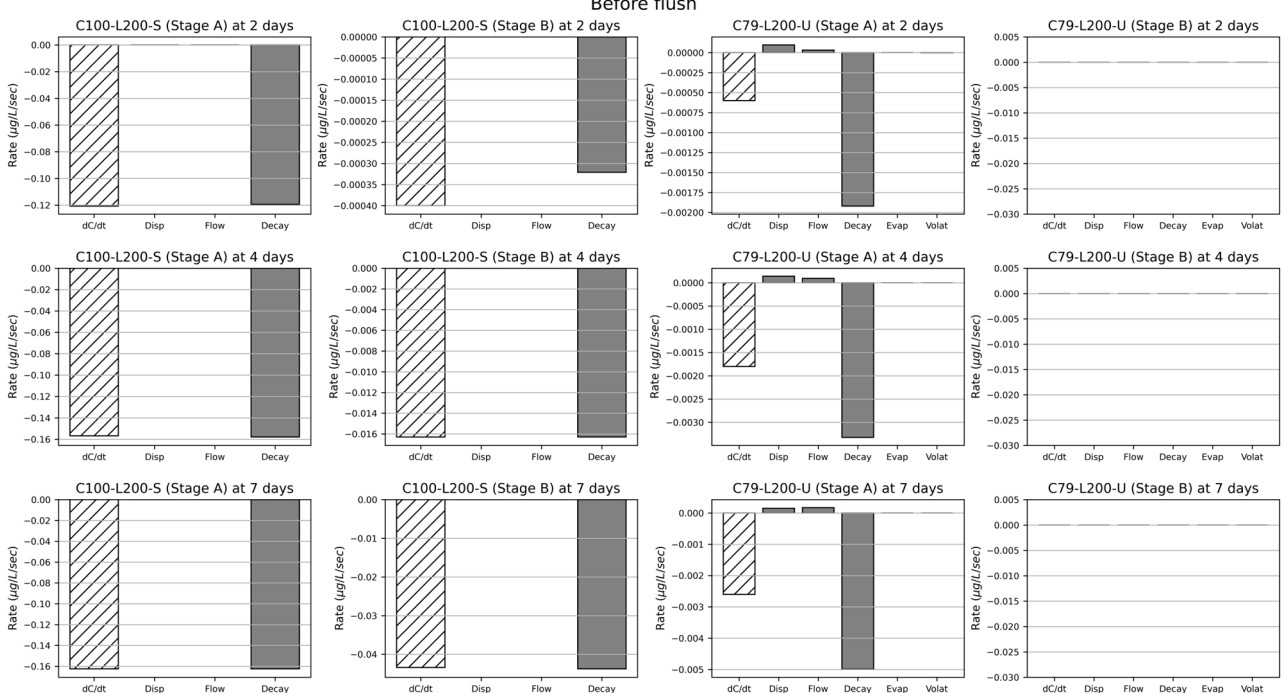

**Figure 7.** Contribution of the different processes (denoted as Disp, Flow, Decay, Evap, Volat) snapshotted at different times (2, 4, 7 days) before a system flush, for selected cases. The leftmost bar denoted as dC/dt indicates the concentration change at either 2, 4 or 7 days: negative values imply a net removal, while positive values imply an accumulation inside the vertical flow CW system.

For carbendazim, the main removal process was consistently Decay at three different time points (after 2, 4, 7 days) for both Stages A and B. Contributions of Disp and Flow were negligibly small. In the case of chlorothalonil, before a system flush, Decay showed the highest contribution to chlorothalonil transport at three different time points (after 2, 4, 7 days) for Stage A. On the other hand, the main contributors in chlorothalonil transport after a system flush were Disp and Flow at three different time points (after 2, 4, 7 days) for Stage A. Contributions of Evap and Volat remained very small over time. Since chlorothalonil was mostly removed at Stage A, the contribution of every process was virtually zero at Stage B. The results indicate that carbendazim seems to have a net removal throughout the CW operation time. However, chlorothalonil seems to accumulate after a system flush before the system transitions to a net removal phase up until the next system flush.

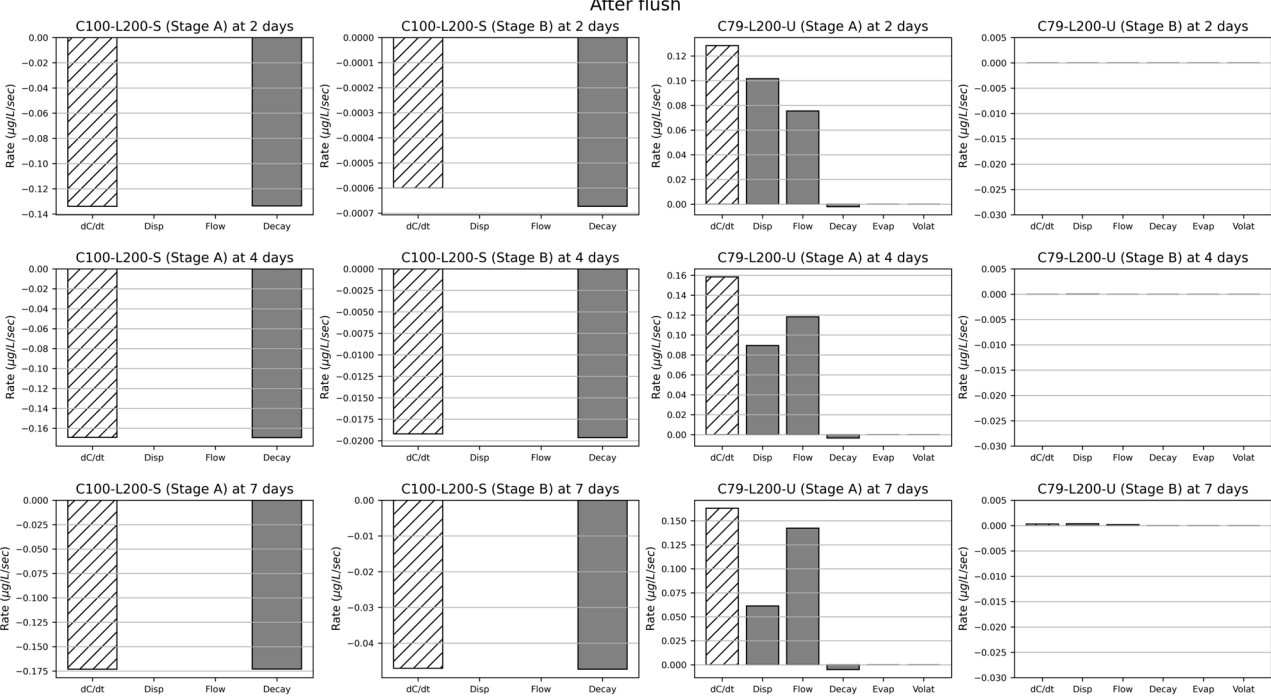

**Figure 8.** Contribution of the different processes (denoted as Disp, Flow, Decay, Evap, Volat) snap-shotted at different times (2, 4, 7 days) after a system flush, for selected cases. The leftmost bar denoted as dC/dt indicates the concentration change at either 2, 4 or 7 days: negative values imply a net removal, while positive values imply an accumulation inside the vertical flow CW system.

### 3.5. Simulation of CW Performance at Standardized Influent Pesticide Concentrations

Once the model was calibrated, simulations were conducted with standardized influent pesticide concentrations. Calculations of the normalized factor were conducted as well to aid in further analysis of the simulated data. The results are as listed in Table 4.

**Table 4.** Simulation data for specific influent pesticide concentrations at different HLRs.

| Condition | Target Pesticide | Influent Concentration ($\mu g\ L^{-1}$) | HLR ($L\ d^{-1}\ m^{-2}$) | Stage A Effluent ($\mu g\ L^{-1}$) | Stage B Effluent ($\mu g\ L^{-1}$) | Normalized Factor |
|---|---|---|---|---|---|---|
| 1 | Carbendazim | 10 | 100 | 53.64 | 1.05 | 0.105 |
| 2 | Carbendazim | 10 | 200 | 26.9 | 2.41 | 0.241 |
| 3 | Carbendazim | 10 | 400 | 22.94 | 3.51 | 0.351 |
| 4 | Carbendazim | 100 | 100 | 536.41 | 10.5 | 0.105 |
| 5 | Carbendazim | 100 | 200 | 269 | 24.12 | 0.2412 |
| 6 | Carbendazim | 100 | 400 | 229.36 | 35.13 | 0.3513 |
| 7 | Chlorothalonil | 100 | 50 | 0 | 0 | 0 |
| 8 | Chlorothalonil | 100 | 200 | 4.85 | 0 | 0.0485 |
| 9 | Chlorothalonil | 100 | 400 | 19.5 | 0 | 0.195 |
| 10 | Chlorothalonil | 500 | 50 | 0.01 | 0 | 0.00002 |
| 11 | Chlorothalonil | 500 | 200 | 24.26 | 0 | 0.04852 |
| 12 | Chlorothalonil | 500 | 400 | 97.48 | 0 | 0.19496 |

The results showed that carbendazim and chlorothalonil removal was dependent on the HLR (carbendazim: F = 2606401, df = 2, and $p = 3.84 \times 10^{-7}$; chlorothalonil:

F = 34343896, df = 2, and $p$ = 2.91 $\times$ $10^{-8}$), but it was independent of the influent concentration (carbendazim: F = 3.57, df = 1, and $p$ = 0.199; chlorothalonil: F = 1.16 $\times$ $10^{-8}$, df = 1, and $p$ = 0.99). The $\eta p^2$ value for effect of HLR on carbendazim and chlorothalonil transport was 0.99 for both the pesticides. This implies that HLR has a large effect on the overall pesticide transport through the CW.

*3.6. Discussion of Results*

This work contributes to a better understanding of pesticide dynamics under various operating conditions (e.g., hydraulic loading rate, influent pesticide concentration, and water saturation) for the vertical flow CW and demonstrated that modeling is a practical tool to assess and predict the performance of the CW with detailed spatial and temporal information. The developed model, in contrast to black-box models, considers the key physicochemical processes behind pesticide removal in the CW, including a hydraulic model to account for the varying water saturation. The model was premised upon and calibrated using experimental results for single-solute CW runs. For this reason, this model is limited to the simulation of a single solute. Apart from this, the model assumes that the solute transport in the modelled system follows a one-dimensional vertical flow process.

The model was tested on the pilot-scale vertical flow CW and simulated the fate of two different pesticides (i.e., carbendazim and chlorothalonil). The key process was a biodegradation process, which is considered important for nature-based environmental technologies including the CW, as it leads to mass depletion of pesticides [34]. The log $K_{ow}$ and $K_{oc}$ affect the sorption process considerably, since they regulate the distribution, mobility and availability of pesticides in different compartments [35]. The target pesticides have a moderate hydrophobic property (2 < log $K_{ow}$ < 3) [13,18]. Chlorothalonil is considered as lowly mobile ($K_{oc}$ > 500) [36]; hence, it is easily adsorbed onto the substrate, while carbendazim is more mobile and continuously transported inside the system. The susceptibility to sorption is also influenced by a molecular structure of pesticide, depending on whether there is a nonpolar region and/or charged groups (i.e., cationic and anionic). Special attention should be paid to strongly sorbing pesticides in the CW, because the sorption capacities will reach the limit at some point and compounds may be released back into the water phase [37].

Considering that the system intends to be used as an apparatus to treat a bulk of wastewater, realization of a higher removal rate with a higher loading rate is preferable from the standpoints of the operation, although we need to be aware of the limit on the loading rate. Higher load of feed water might have caused disappearance of heterogeneity, resulting in larger factor values associated with the distribution coefficient and degradation rate constant than those for lower load, indicating that an enhanced removal occurred under our field condition compared to experiments of previous studies. This was in agreement with a recent study [35], which proposed site-specific modulating factors for achieving more accurate simulations. This factor can also include the concept of temporal variations of process efficiencies (e.g., plant growth and biofilm development) inside the system occurring as the treatment proceeds. Our results indicate that a higher carbendazim removal rate was obtained during the saturated condition, partially due to a prolonged retention time. Long periods of stagnation could promote solute contact with the substrate and may have facilitated the removal of contaminants from the water. As an alternative operation, alternating drainage with non-drainage periods in the CW could be worth applying, since this will foster aerated conditions to support aerobic bacteria, boosting degradation of the compound further [37]. Adhikary et al. (2022) [38] obtained removal rates of up to 99.6% for carbendazim using a laboratory-scale vertical flow CW. However, they set pH at 2 to facilitate adsorption, and the removal rates considerably dropped to neutral to alkaline pH conditions.

The robustness of the developed model will be reinforced, testing with various types of pesticides with different hydrophobicity, mobility and volatility, as well as under different field climate conditions. Due to the nature of mechanistic model, the developed model is

readily configurable using existing chemical property values that can be retrieved from the literature and will become a versatile tool in terms of the applicability to other pesticides. We did not explicitly include the role of plants in the dissipation of pesticide. However, plants indeed affect the fate of pesticide, complicating water flow paths, attracting microorganisms responsible for pesticide transformation, contributing to the increase in the organic carbon content and composition diversity to enhance adsorption, and bringing oxygen to the bed to enhance aerobic reactions [35]. The pH, temperature and dissolved oxygen content are also important factors among physicochemical parameters for pesticide removal processes [34]. In addition, nutrient (N and P) transformation cycle affects activities of microorganisms involving pesticide transformation. The inclusion of those elements into the model will be beneficial. Furthermore, the model can be extended to a vertical 2D (x,z) model that could make it possible to analyze heterogeneities (e.g., dead zones) existing in the lateral (x) direction. Dispersion will govern the water transport inside dead zones, while advection governs inside the main channel, resulting in heterogeneous distribution of dissolved pesticide concentrations in the CWs [39].

## 4. Conclusions

A mechanistic model for single-solute one-dimensional transport based on a mass balance approach was developed in this study. This model was designed to simulate the pesticide transport through a two-stage vertical flow constructed wetland design, which has been previously used for experimental studies of carbendazim and chlorothalonil removal from synthetic wastewater.

A good calibration of the model was achieved based on the corresponding experimental results for each of the initial simulation cases. As part of the calibration process, two new parameters of mass transfer resistance and enhancement of degradation were introduced to the simulation calculations. The introduction of these two parameters was a meaningful addition to the simulation process, and both these parameters should be explored in further detail.

The simulation of carbendazim and chlorothalonil transport at specific influent concentrations and HLRs showed an accumulation of pesticide in the substrate bed over time. This led to a gradual increase in the effluent pesticide concentrations until an almost steady-state condition was reached after a week of system operation. The simulated effluent pesticide concentrations were positively correlated with the HLR of the influent and independent of the influent pesticide concentrations, which were consistent with the experimental findings.

An analysis of the individual processes considered in the simulation model for carbendazim showed that degradation was the predominant process throughout the simulation period. The simulation model for chlorothalonil showed that the predominant process was initially advective flow and then degradation.

The developed model is not limited to simulating only carbendazim and chlorothalonil transport through the CW. The solute transport for any specific pesticide can be modelled once the values of chemical properties used in the model are updated to those of the desired pesticide. The substrate physical characteristics can also be updated within the code, to further explore the effect of different substrate characteristics on the pesticide transport through the CW.

The model developed in this study can find potential application in the floriculture industry, where it can be an important tool for the design and construction of treatment wetlands for pesticide removal from wastewater. This study can also be considered as a good base for the development of more complex solute transport simulations, such as multiple-solute transport and two-dimensional solute transport.

**Supplementary Materials:** The following are available online at https://www.mdpi.com/article/10.3390/w16010142/s1, File S1: This is an f90 file with the Fortran code used for simulation calculations. File S2: This is a Jupyter notebook file containing the python code used for visualization of the simulated data. Table S3: This table contains values of newly introduced parameters for model calibration for each simulation case and their corresponding error values.

**Author Contributions:** Conceptualization, S.W. and Y.I.; methodology, S.W.,Y.I. and A.D.; software, Y.I. and A.D.; validation, Y.I. and F.Z.; formal analysis, S.W., Y.I. and A.D.; investigation, S.W. and Y.I.; resources, Y.I. and T.U.K.N.; data curation, S.W., Y.I. and A.D.; writing—original draft preparation, S.W.; writing—review and editing, Y.I., F.Z., T.U.K.N., W.S. and A.D.; visualization, Y.I.; supervision, Y.I. and F.Z.; project administration, S.W.; funding acquisition, S.W. All authors have read and agreed to the published version of the manuscript.

**Funding:** This research received no external funding.

**Data Availability Statement:** The data presented in this study are available in the Supplementary Materials provided alongside.

**Acknowledgments:** The authors would like to thank Daiki Saito for his assistance in the early stages of the model development. We are also grateful to the anonymous peer reviewers for their insightful comments and suggestions. Their contributions have helped improve the coherence and quality of this paper.

**Conflicts of Interest:** The authors declare no conflicts of interest.

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
