# Peer review of "A Mechanistic Model for Simulation of Carbendazim and Chlorothalonil Transport through a Two-Stage Vertical Flow Constructed Wetland"

_water, doi:10.3390/w16010142_

Round 1
Reviewer 1 Report
Comments and Suggestions for Authors
Water-MDPI: A mechanistic model for simulation of carbendazim and chlorothalonil transport through a two-stage vertical flow constructed wetland.
The following comments need to be addressed.
· Minimize abbreviations in the abstract. Several abbreviations are used.
· Define abbreviations at their first appearance in the main sections (Introduction section onwards). Do not use the abbreviation after defining it one time in the abstract – it must be defined in the main texts. Avoid one-time or two-times used abbreviations. Few abbreviations are used without defining them.
· “Error! Reference source not found” is found in 13 places. Rectify it!
· It is a poorly written manuscript with a lack of recent literature (very few referred from 2021 to 2023).
· Identify the proper research gap from the recent literature (2022 & 2023).
· Web references are about 20% of the list of references. Enrich the literature section with the most relevant information from recent literature.
· What parameters were introduced during the calibration process of the mechanistic model for single solute one-dimensional transport, and why were they considered meaningful additions to the simulation calculations?
· Describe the observed trend in the simulated effluent pesticide concentrations over time in the two-stage vertical flow constructed wetland design. How does this trend correlate with the hydraulic loading rate (HLR) of the influent and the influent pesticide concentrations?
· In the simulation model for carbendazim, which process was identified as the predominant one throughout the simulation period, and how does this finding contribute to understanding pesticide transport in the constructed wetland?
· Contrast the predominant processes identified in the simulation models for carbendazim and chlorothalonil. How do these processes vary over the simulation period, and what implications do they have for pesticide removal in the constructed wetland?
· How versatile is the developed model in simulating solute transport through the constructed wetland?
· Explain how the model can be adapted to simulate the transport of specific pesticides other than carbendazim and chlorothalonil, and what considerations are involved in updating the model for different chemical properties and substrate characteristics.
· Figures lack clarity and resolution. Provide high-resolution images.
· Provide equation numbers.
· Provide proper source citations for equations.
Comments on the Quality of English LanguageMinor editing of English language required
Author Response
Reviewer #1
We would like to thank you for your comments on our manuscript. We have worked towards improving the manuscript with your comments and suggestions in mind. Please find a detailed explanation of the changes made in response to the same. We hope that our revisions are satisfactory.
Comment No. 1: Minimize abbreviations in the abstract. Several abbreviations are used.
We have removed abbreviations from the abstract in response to this suggestion.
Comment No. 2: Define abbreviations at their first appearance in the main sections (Introduction section onwards). Do not use the abbreviation after defining it one time in the abstract – it must be defined in the main texts. Avoid one-time or two-times used abbreviations. Few abbreviations are used without defining them.
This suggestion followed and changes were implemented throughout the manuscript document.
Comment No. 3: “Error! Reference source not found” is found in 13 places. Rectify it!
There had been an error in the cross-references to tables and figures post-submission, when the manuscript was typeset. These errors have now been fixed.
Comment No. 4: It is a poorly written manuscript with a lack of recent literature (very few referred from 2021 to 2023).
Comment No. 5: Identify the proper research gap from the recent literature (2022 & 2023).
In line with comments no. 4 and 5, additions have been made to the introduction section (lines 83-97) based on recent literature. Furthermore, a new discussion section has been included after the results to evaluate the results as a whole (section 3.5, line 460).
Comment No. 6: Web references are about 20% of the list of references. Enrich the literature section with the most relevant information from recent literature.
The number of web references have been reduced. The literature section has been enriched with recent literature as well as reliable sources of data.
Comment No. 7: What parameters were introduced during the calibration process of the mechanistic model for single solute one-dimensional transport, and why were they considered meaningful additions to the simulation calculations?
In response to this comment, we have moved the description of the new calibration parameters from the results section to the simulation methods section (lines 303-313).
Comment No. 8: Describe the observed trend in the simulated effluent pesticide concentrations over time in the two-stage vertical flow constructed wetland design. How does this trend correlate with the hydraulic loading rate (HLR) of the influent and the influent pesticide concentrations?
The two-factor ANOVA tables were removed and the results were stated in text. The effect size of HLR on pesticide transport was evaluated by means of partial eta-squared calculation. The implications of the calculated partial eta-squared values were also mentioned.
Comment No. 9: In the simulation model for carbendazim, which process was identified as the predominant one throughout the simulation period, and how does this finding contribute to understanding pesticide transport in the constructed wetland?
Comment No. 10: Contrast the predominant processes identified in the simulation models for carbendazim and chlorothalonil. How do these processes vary over the simulation period, and what implications do they have for pesticide removal in the constructed wetland?
In line with comment no. 9 and 10, the contributions of individual processes were analyzed again in further detail. This section was rewritten in a more comprehensive manner to reflect the new findings. Further points were elaborated on in the discussion section to relate the findings to recent literature.
Comment No. 11: How versatile is the developed model in simulating solute transport through the constructed wetland?
Comment No. 12: Explain how the model can be adapted to simulate the transport of specific pesticides other than carbendazim and chlorothalonil, and what considerations are involved in updating the model for different chemical properties and substrate characteristics.
In line with comment no. 11 and 12, the versatility of the model was elaborated in the new discussion section. Details of how the model can be adapted to simulating transport of other pesticides in a CW were added as well.
Comment No. 13: Figures lack clarity and resolution. Provide high-resolution images.
All the images have been replaced with high resolution images.
Comment No. 14: Provide equation numbers.
Comment No. 15: Provide proper source citations for equations.
In line with comment no. 11 and 12, the reference for the equations has been added in the methods section. Equation numbers have been added as well.
Reviewer 2 Report
Comments and Suggestions for Authors
The paper was interesting and the results are generally well presented. I have listed some comments for consideration, which I hope the authors find helpful.
1. Introduction. Please include some reliable references to ensure that the statistics (e.g., $421 million) are reliable in the opening paragraph.
2. What is the model development on line 77? I think slightly more context is needed here for the reader to understand what this would entail (e.g., a prediction model that is able to accurately predict samples and expedite the sampling process?) The next sentence goes into some detail about this, but it not clear what is being predicted or optimized.
3. Please move lines 82-86 to the next section. I think this information can be merged with the existing paragraph in a way that makes the overall method clearer.
4. Please check the unfortunate “Error” references. These appear in numerous pleases across the paper.
5. Section 2.1 is generally unclear. How are the cases simulated (e.g., computer-based simulation or theoretical calculation?) I think it is the latter given the information provided in the following section, which suggests that the information needs to be reorganized.
6. Table 2. Why are there two references for the same row? It is unclear what is being cited.
7. Section 2.6.1 this information belongs in the discussion section.
8. I am not sure whether Table 5 is necessary. Would it be simpler to include the results of the ANOVA in the text, by only including the essential statistical values (i.e., F-value, df, and p-value.) I would also recommend including the effect size (e.g., partial eta-squared.) Please also format the values consistently.
9. There is no discussion section.
Author Response
Reviewer #2
We would like to thank you for your comments on our manuscript. We have worked towards improving the manuscript with your comments and suggestions in mind. Please find a detailed explanation of the changes made in response to the same. We hope that our revisions are satisfactory.
Comment No. 1: Introduction. Please include some reliable references to ensure that the statistics (e.g., $421 million) are reliable in the opening paragraph.
New reliable references from government source were used for statistics.
Comment No. 2: What is the model development on line 77? I think slightly more context is needed here for the reader to understand what this would entail (e.g., a prediction model that is able to accurately predict samples and expedite the sampling process?) The next sentence goes into some detail about this, but it not clear what is being predicted or optimized.
In line with this comment, clarifications were made with regards to the type of simulations referred to in this sentence.
Comment No. 3: Please move lines 82-86 to the next section. I think this information can be merged with the existing paragraph in a way that makes the overall method clearer.
These relevant lines were moved to the methods section under a new subsection of “Model description”.
Comment No. 4: Please check the unfortunate “Error” references. These appear in numerous pleases across the paper.
There had been an error in the cross-references to tables and figures post-submission, when the manuscript was typeset. These errors have now been fixed.
Comment No. 5: Section 2.1 is generally unclear. How are the cases simulated (e.g., computer-based simulation or theoretical calculation?) I think it is the latter given the information provided in the following section, which suggests that the information needs to be reorganized.
The simulation section was moved to subsection 2.6.1, as part of the simulation methods. The change should reflect how computer-based simulations are used in our study, and not theoretical calculations.
Comment No. 6: Table 2. Why are there two references for the same row? It is unclear what is being cited.
The table in question is now numbered Table 1, and has been reformatted to better present the intended data in an organized manner without any misrepresentation.
Comment No. 7: Section 2.6.1 this information belongs in the discussion section.
The said section has been removed and has been included in the new discussion section after the results (line 467).
Comment No. 8: I am not sure whether Table 5 is necessary. Would it be simpler to include the results of the ANOVA in the text, by only including the essential statistical values (i.e., F-value, df, and p-value.) I would also recommend including the effect size (e.g., partial eta-squared.) Please also format the values consistently.
In line with this comment, Table 5 and 6 have been removed. Instead, ANOVA results have been included in the text. The effect size of HLR was calculated and discussed as well.
Comment No. 9: There is no discussion section.
A new discussion section (section 3.5, line 460) has been included in the manuscript. Results have been evaluated as a whole and correlated with recent literature.
Round 2
Reviewer 1 Report
Comments and Suggestions for Authors
The revised version is made satisfactorily.
Comments on the Quality of English LanguageMinor editing is required.
Reviewer 2 Report
Comments and Suggestions for Authors
Thank you for addressing all my previous comments. I have one suggestion below, which is very minor. Please change "partial n2" to "np2."